# Natural Disasters as a Maternal Prenatal Stressor and Children’s Neurodevelopment: A Systematic Review

**DOI:** 10.3390/bs14111054

**Published:** 2024-11-06

**Authors:** Gül Ünsel-Bolat, Sema Yıldırım, Fethiye Kılıçaslan, Rafael A. Caparros-Gonzalez

**Affiliations:** 1Department of Child and Adolescent Psychiatry, Balıkesir University, 10145 Balıkesir, Turkey; gul.unsel.bolat@gmail.com; 2Department of Child and Adolescent Psychiatry, Harran University, 63300 Şanlıurfa, Turkey; fethiyeklcaslan@gmail.com; 3Faculty of Health Sciences, University of Granada, 18071 Granada, Spain; rcg477@ugr.es; 4Instituto de Investigacion Biosanitaria ibs.GRANADA, 18071 Granada, Spain

**Keywords:** prenatal maternal stress, natural disaster, child neurodevelopment

## Abstract

The intrauterine period is a time of high sensitivity in the development of the embryo and the fetus. Therefore, low levels of maternal stress are closely associated with healthy brain development in the neonatal and early childhood periods. There is increasing evidence linking natural disasters as prenatal maternal stress (PNMS) to neurodevelopmental disorders (including subclinical manifestations). Natural disasters involve many factors in addition to the trauma they cause, including loss and the physical and psychosocial difficulties that result from that trauma. This review article aims to bring together research findings on the neurodevelopmental effects of natural disasters on children as PNMS. It also looks at how factors such as gestational age and gender contribute to these effects. We conducted a systematic review on PubMed, Web of Science, and Scopus, with 30 studies meting the inclusion criteria. This systematic review was conducted in accordance with the PRISMA guidelines. A total of 1,327,886 mother–child dyads participated in the included studies. The results of the studies indicate that natural disasters have a negative impact on children’s outcomes in terms of cognitive development, language development, autism/autism-like features, motor skills, performance in mathematics, mental development, sleep, attention, behavioral and emotional problems, and various psychiatric comorbidities.

## 1. Introduction

Worldwide, more than 400 natural disasters occur every year. A natural disaster is defined as a natural hazard event causing an economic loss of at least 50 million U.S. dollars or 2000 homes damaged and include floods, earthquakes, fires, or hurricanes [1,2]. Natural disasters can be considered nature experiments since a significant number of people are exposed to the same event, occurs in a specific period of time, and has a high stress load. This approach is analogous to a controlled scientific experiment design, as the level of exposure to the stressful situation is largely independent of factors such as socioeconomic status, personality, and genotype, which can be confounded with other types of stressful life events [3]. However, children and families from disadvantaged groups may be more vulnerable to disasters due to their social position [4]. For example, low-income communities, people with disabilities, ethnic and racial minorities, and families living in poorly constructed houses or with limited access to food supplies are more likely to be affected by an earthquake than families with more resources.

During a natural disaster, certain intrauterine negative consequence may arise (low birth weight, still birth) [5]. The intrauterine period is a time of high sensitivity in the development of the embryo and fetus, and highly stressful situations such as natural disasters can have detrimental consequences on offspring [6,7]. Pregnant women and neonates are considered a vulnerable population to natural disasters [8,9,10]. In addition, natural disasters can result in the premature rupture of membranes, preterm labor, low birth weight, and infectious diseases [11,12,13]. Additionally, the challenges of women’s working lives may make them more vulnerable to disaster-related economic impacts, resulting in reduced access to health resources [8]. Furthermore, pregnant women exposed to natural disasters are vulnerable to a higher prevalence of mental health problems (anxiety, depression) [14]. Perceived stresses such as pregnant women’s worries about their babies and their own health, changes in physical activity due to pregnancy, and postpartum parenting concerns are significantly associated with both depression and post-traumatic stress disorder [15]. Therefore, pregnant women are more likely to experience both physical and mental health issues during and after a natural disaster compared to the general population [13].

Prenatal maternal stress (PNMS) is defined as a pregnant mother’s experience of distress due to various environmental exposures and their associated affective response to those stressors [16]. Maternal psychopathology and stress during pregnancy are among the most common intrauterine exposures that negatively affect the health of the fetus in the prenatal period [17,18]. This has been associated with preterm birth and low birth weight [19]. In the long term, it has been associated with behavioral, emotional, cognitive, and motor problems in childhood [20,21] and psychiatric disorders among adolescents [22,23]. High levels of maternal stress are closely associated with brain development delays in neonates and toddlers [7,24].

Natural disasters, in addition to the trauma they create, also include many factors, including losses and physical and psychosocial difficulties that develop due to this trauma. According to our research, there are only a few studies examining various problems in the children of women exposed to natural disasters during pregnancy. Along with this limited number of studies, the number of reviews on this subject is also quite limited. The reviews we examined have some limitations; in one review, only the mental problems of pregnant women exposed to an earthquake were mentioned, and the effects on children were not evaluated [25]. In a review, only the effects of flood natural disasters were examined [26], and in a meta-analytical review examining the comprehensive effects of natural disasters on children as PNMS, only ice storm, flood, and cyclone effects were studied as meta-regression [27]. In an additional review conducted in 2020, Blanc et al. examined only studies on the 2010 Haiti Earthquake [28]. In the few reviews conducted so far, mostly one type of natural disaster has been evaluated. In light of all these data, we planned to conduct a study that covers all types of natural disasters and their neurodevelopmental effects in offspring. In addition, we aimed to evaluate the effect of gestational age and sex, which have not been evaluated sufficiently before.

For this reason, the aim of this study was to investigate the neurodevelopmental effects in children exposed to natural disasters during the intrauterine period. Furthermore, the effect of gestational age and sex of the child on these neurodevelopmental changes was also assessed. The aim of this review was to systematically summarize the research findings and to facilitate collaboration between related disciplines.

## 2. Materials and Methods

This review was conducted in accordance with the PRISMA 2020 guidelines [29]. This review was registered in Open Science Framework with registration Doi: https://doi.org/10.17605/OSF.IO/5CQX6 (accessed on 10 September 2024). PRISMA 2020 expanded checklist can be found in Appendix A. The inclusion and exclusion criteria can be reviewed in Table 1.

### 2.1. Literature Search and Selection Studies

Searches were conducted between May and September 2024 on PubMed, Web of Science, and Scopus databases by 2 independent searchers (S.Y. and G.Ü.-B.). The search strategy included keywords related to natural disasters, prenatal maternal stress, and child neurodevelopment. The search terms used on the databases were as follows: (“natural disasters” OR “disasters” OR “earthquake” OR “flood” OR “tornado” OR “hurricane” OR “storm” AND “prenatal maternal stress” OR “antenatal stress” OR “maternal stress” OR “pregnancy” OR “prenatal exposure delayed effects” AND “neurodevelopment disorders” OR “language development” OR “motor development “OR “cognitive development” OR “adhd” OR “autism spectrum disorder” AND “children” OR “offspring” OR “adolescents”). A total amount of 1971 records were identified. After full-text review, 30 articles met the inclusion criteria.

### 2.2. Data Collection and Analyses

Eligible studies were selected through a multistep approach (elimination of duplicates, title reading, abstract, and full-text assessment). Two researchers (G.Ü.-B. and S.Y.) collaboratively screened the titles and abstracts, followed by a thorough evaluation of the full texts based on the inclusion criteria outlined above. Any disagreement between the reviewers was solved by means of a consensus session with a third reviewer (F.K.). The entire process was supervised by R.A.C.-G. In the case of ambiguity in reporting or lack of data, primary authors were contacted for clarification. Sixty-four were potentially eligible for inclusion based on the title and abstract. After full-text review, thirty articles met the inclusion criteria. The PRISMA 2020 [29] flow diagram summarizes the study selection process (Figure 1).

### 2.3. Data Extraction and Management

The data were extracted by two researchers (G.Ü.-B. and S.Y.). During the assessment process, both authors reviewed all studies together, agreed upon the assessment criteria, and considered the following information for each article: (1) first author and year of publication; (2) natural disaster and the country where the disaster occurred; (3) study design; (4) the assessment tool used in every study; (5) number of participants; (6) the average age of infants population; (7) the gestational age of pregnant women; (8) main results obtained; and (9) the most important findings of each study.

### 2.4. Quality Assessment Tools

The quality of each cohort and cross-sectional study was assessed using the Newcastle–Ottawa Quality Assessment Scale (NOQAS) [30]. The quality of each natural experiment and quasi-experimental study was assessed using the TREND Statement Checklist [31].

## 3. Results

### 3.1. Study Characteristics

A total of 30 articles written in English were included in this review (Table 2). The countries where natural disasters occurred were Australia (*n* = 11), Canada (*n* = 5), China (*n* = 3), Chile (*n* = 3), India (*n* = 2), and the USA (*n* = 6). The type of natural disasters included earthquakes (*n* = 4), floods (*n* = 13), hurricanes (*n* = 4), ice storms (*n* = 5), sand storms (*n* = 1), cyclones (*n* = 2), and others (a combination of hurricanes, tornadoes, floods, storms) (*n* = 1). Twenty-one studies were cohort studies, and there was one cross-sectional study, five natural experiment studies, and three quasi-experimental studies. A total of 1,327,886 mother–child dyads participated in the included studies.

### 3.2. Maternal Stress and Socio-Emotional Development in Offspring

Eleven of the reviewed studies [32,33,34,35,36,37,38,39,40,41,42] indicate that maternal stress due to exposure to natural disasters during pregnancy negatively affects the emotional development of children. The scales used among the studies were the CBCL (Child Behavior Checklist), QFOSS (Queensland Flood Objective Stress Scale), PDI (Peritraumatic Distress Inventory), PDEQ (Peritraumatic Dissociative Experiences Questionnaire), IES-R (Impact of Event Scale—Revised), DASS-21 (Depression Anxiety Stress Scales-21), BDI (Beck Depression Inventory), TADI (Teacher’s ADHD Rating Scale), LES (Life Events Scale), STAI (State-Trait Anxiety Inventory), BITSEA (Brief Infant–Toddler Social and Emotional Assessment), SPAS (Spence Preschool Anxiety Scale), IBQ-R (Infant Behavior Questionnaire—Revised), PAPA (Preschool Age Psychiatric Assessment), and ECBQ (Early Childhood Behavior Questionnaire).

It was observed that the major objective difficulties experienced by pregnant women exposed to the Queensland floods as a natural disaster during pregnancy were associated with higher anxiety symptoms in their 4- and 6-year-old children, and that negative reactivity and timid temperament traits in the toddler age group mediated the relationship between these anxiety symptoms [33,34,35,36]. It has even been shown that these anxiety symptoms at 2.5 years of age predict sleep problems (sleep duration, awakenings) at 4 years of age [40]. Children exposed in utero to the Chilean earthquake had more negative socio-emotional outcomes in early childhood [32], with scores on behavioral problems such as emotional reactivity, anxious/depressed mood, sleep and attention problems, and aggression being significantly higher in these children than in children not exposed to the earthquake [37].

Laplante et al. concluded that toddlers of pregnant women exposed to the Quebec ice storm engaged in less functional play and more stereotypical play [42]. In studies related to Hurricane Sandy, prenatal maternal depression has been shown to affect infant temperament, including poorer emotion regulation and increased negative affect, distress, and sadness in infants [38,41]. A significant increase in the risk of various psychiatric disorders, including anxiety disorders, depressive disorders, and attention deficit disorder/disruptive behavior disorder, has been observed in preschool children exposed to the same natural disaster in the womb [39].

### 3.3. Exposure to Natural Disasters and Infants/Toddlers/Children’s Neurodevelopment

In most of the studies [32,43,44,45,46,47,48,49,50,51,52,53,54,55,56,57,58,59,60,61], neurodevelopmental effects in children of pregnant women exposed to natural disasters as prenatal maternal stress have been examined. Therefore, more data were obtained on this subject. The MCDI-III (MacArthur–Bates Communicative Development Inventories-III), TADI (Teacher’s ADHD Rating Scale), CBCL (Child Behavior Checklist), DST (Developmental Screening Test), BOTMP (Bruininks–Oseretsky Test of Motor Proficiency), VMI (Visual–Motor Integration), BOT-2 (Bruininks–Oseretsky Test of Motor Proficiency, Second Edition), Gesell DI (Gesell Developmental Inventory), WISC-CR (Wechsler Intelligence Scale for Children—Chinese Revised), Conner’s TRS-R (Conners’ Teacher Rating Scale—Revised), ASQ (Ages and Stages Questionnaire), Bayley Scales of Infant Development-MDI (Mental Development Index), MCDI (MacArthur–Bates Communicative Development Inventories), WPPSI-R (Wechsler Preschool and Primary Scale of Intelligence—Revised), PPVT-R (Peabody Picture Vocabulary Test—Revised), BITSEA (Brief Infant–Toddler Social and Emotional Assessment), and PAPA (Preschool Age Psychiatric Assessment) scales were used to evaluate neurodevelopmental effects.

Twenty studies have shown that in utero exposure to natural disasters negatively affects children’s neurodevelopment [32,43,44,45,46,47,48,49,50,51,52,53,54,55,56,57,58,59,60,61]. These neurodevelopmental effects in studies include cognitive development, language development, reading and pre-reading skills, fine and gross motor skills, mathematical skills, attention deficit hyperactivity disorder symptoms, and autism spectrum disorder symptoms.

In the Developmental Screening Test (DST) conducted on children aged 0–3 years of 89 mothers exposed to the Wenchuan earthquake during pregnancy, higher maternal PTSD (post-traumatic stress disorder) scores were significantly associated with lower child Developmental Quotient (DQ) and Mental Index (MI) scores (*p* < 0.01) [45]. The pre-reading skills of kindergarten-aged children of pregnant women exposed to the Chilean earthquake were found to be lower than those of the nonexposed group (*p* < 0.001) [44]. The neurodevelopmental effects of natural disasters are not limited to the preschool age group. A study of third grade standardized math and reading test scores of 187,000 children aged 8–10 who were exposed in utero to 15 different natural disasters in the US between 1988 and 2000, and 693,967 children who were not exposed in utero, showed that the exposed group had lower math (*p* < 0.01) and reading (*p* < 0.05) scores [47]. Other studies have shown that cognitive effects are significant [32,43,49,51,53,54,55,57].

A study examining the motor development of children of pregnant women exposed to the 2011 Queensland floods found that high prenatal maternal stress was positively associated with motor development in 2-month-old children but negatively associated with motor development in 6- and 16-month-old children, particularly when exposure occurred late in pregnancy [59]. A study of the same natural disaster concluded that high levels of maternal PTSD symptoms were associated with poorer fine motor skills in children (*p* < 0.05) [58]. Gomula et al. showed that prenatal exposure to the 2009 Cyclone Aila disaster was associated with poorer performance on motor skills tests (*p* < 0.05) [48] in a study conducted with prenatal (*n* = 290), postnatal (*n* = 169), and nonexposed (*n* = 260) groups.

Studies have also mentioned the negative impact of natural disaster exposure on language development [43,53,54,56]. Weaker language skills were observed in children exposed to high levels of PNMS compared to those exposed to moderate or low levels of PNMS caused by natural disaster exposure [54]. Li et al. reported that every 10 days of prenatal exposure to PNMS caused a 0.20 SD decrease in vocabulary test scores, leading to delays in sentence construction and speech [56].

In addition, there are studies showing that children exposed to natural disasters as PNMS have higher scores on autism spectrum disorder screening tests [61] and even an increased prevalence of ASD [52] and higher symptoms of attention deficit hyperactivity disorder [50].

### 3.4. Exposure of Pregnant Women to Natural Disasters: Implications of Trimester of Pregnancy and Children’s Gender

In these studies [34,42,44,48,49,51,52,53,55,56,58,59,61], researchers not only examined the neurodevelopmental effects of exposure to natural disasters in children but also whether the period of pregnancy during which exposure to natural disasters occurs is effective or not. Most of the studies reviewed found that exposure to natural disasters in the first trimester resulted in worse neurodevelopmental effects (poorer reading skills, cognitive development, intellectual abilities, language abilities, functional play, motor development, presence of symptoms of autism spectrum disorder) in children [42,44,49,51,53,55,58,61]. While some studies suggest that exposure in late pregnancy leads to worse outcomes [52,56,59], two studies found no significant association between gestational age of exposure and neurodevelopmental effects (motor function, anxiety symptoms) in children (*p* < 0.01) [34,48].

In three studies investigating the impact of gender, while significant results were obtained in terms of hyperactivity, behavioral problems, and poorer fine motor skills in boys [33,39,60], it was mentioned that problems with reading and language skills were more common in girls [44,55].

**Table 2 behavsci-14-01054-t002:** Characteristics of included studies.

First Author, Year	Natural Disaster	Country	Study Design	Assessment Tool	Sample Size	Infants Mean Age	Gestational Age at Birth (Weeks)	Main Results	Conclusions	Quality of the Study *
Austin et al., 2017 [43]	Queensland flood 2011	Australia	Prospective cohort study	Questionnaire: QFOSS, PDI, PDEQ, IES-R, DASS-21, MCDI-III, Bayley-III Scales of Infant and Toddler Development	*n* = 128 mother–child dyads	30 months old (SD:NR)	NR	- Prenatal stress not associated with toddlers’ cognitive and language development at 30 months (*p* > 0.05). - Maternal structuring and sensitivity associated with toddlers´ cognitive outcomes (*p* < 0.05).	Maternal structuring and sensitivity play a significant role in enhancing the language development of children whose mothers experienced high levels of stress during pregnancy.	7/9
Batiz et al., 2021 [44]	Chilian Earthquake 2010	Chile	Retrospective cohort study	Platform; DIALECT (a validated Spanish reading diagnostic instrument)	3280 mother–child dyads (earthquake exposed *n* = 865; nonexposed *n* = 2415)	Kindergarten age	Exposed group; 1st trimester *n* = 292; 2nd trimester *n* = 337; 3rd trimester *n* = 236	- Children exposed to the earthquake demonstrated lower pre-reading skills compared to their unexposed peers (*p* < 0.001). - The most significant impairments were observed in children exposed during the first trimester (*p* < 0.05). Additionally, there were notable sex differences, with certain skills being more affected in females and others in males.	Prenatal exposure to stressful events such as earthquakes can negatively influence early reading skills in children. The timing of exposure, particularly during the first trimester, plays a crucial role in the extent of these effects.	8/9
Berthelon et al., 2021 [32]	Chilian Earthquake 2010	Chile	Cohort study	Questionnaire: BDI, TADI, CBCL	2045 mother–child dyads (earthquake exposed *n* = 727; nonexposed *n* = 1318)	19.4 ± 6.45 months old	1st trimester *n* = 446; 2nd trimester *n* = 726; 3rd trimester *n* = 727	- Cognitive skills were more adversely affected if stress occurred in the first trimester of pregnancy, while socio-emotional behaviors were more negatively impacted if stress occurred in the last trimester (*p* < 0.05).	Prenatal maternal stress due to the earthquake negatively affects early childhood cognitive and socio-emotional outcomes. Prenatal maternal stress has different negative effects depending on the stage of pregnancy.	8/9
Cai et al., 2017 [45]	Wenchuan earthquake 2008	China	Cross-sectional	Questionnaire: PTSD Checklist (PCL-C), DST	89 mother–child dyads	Childs 0–3 years old (SD:NR)	NR	- Higher PTSD scores in mothers correlated with lower Developmental Quotient (DQ) and Mental Index (MI) scores in their children (*p* < 0.01).	Maternal PTSD due to earthquake exposure has a negative impact on the mental development of children, with significant associations between higher PTSD scores in mothers and lower cognitive outcomes in their offspring.	5/9
Cao et al., 2014 [46]	Quebec Ice Storm 1998	Canada	Cohort study	Questionnaire: BOTMP, Beery-Buktenica Developmental Test of VMI, IES-R, LES, EDPS	89 mother–child dyads (*n* = 42 boys; *n* = 47 girls)	5.5 (SD; NR) years old	1st trimester *n* = 35; 2nd trimester *n* = 27; 3rd trimester *n* = 27	- Higher levels of objective and subjective PNMS were associated with lower scores in motor function tasks (*p* < 0.01). - The timing of exposure, sex of the child, and the type of stress (objective or subjective) played a significant role in determining the outcomes. - Girls exposed to stress later in pregnancy showed worse motor function outcomes compared to boys (*p* < 0.05).	Both objective hardship and subjective distress due to prenatal maternal stress have a negative impact on children’s motor functions.	7/9
Fuller S.C., 2014 [47]	15 different natural disasters (hurricanes, tornadoes, floods, and storms) in North Carolina between 1988 and 2000	USA	Cohort study	Standardized Third Grade End-of-Grade Tests in Math and Reading, Special Education and Gifted Placement Rates	880.967 mother–child dyad (exposed kids *n* = 187.000)	Age between 8 and 10 years old (SD; NR)	NR	- This study found that children prenatally exposed to hurricanes had lower math (*p* < 0.01) and reading (*p* < 0.05) scores in third grade, with a significant effect among disadvantaged subgroups, particularly children born to black mothers (*p* < 0.01).	Children prenatally exposed to hurricanes had lower math and reading scores in third grade, with a significant effect among disadvantaged subgroups, particularly children born to black mothers.	8/9
Gomula et al., 2023 [48]	Cyclone Aila 2009	India	Cohort study	Questionnaire: BOT-2	*n* = 719 mother–child dyad (td = 260; prenatal exposed; 290; postnatal exposed; 169)	8.47 ± 0.55 months old	NR	- Prenatal exposure was associated with poorer performance in most motor skills tests, except for fine motor precision and strength (*p* < 0.05). - There were no differences in motor functions relative to the timing of the exposure (trimester) during pregnancy (*p* > 0.05).	Both prenatal and early infancy exposure to severe natural disasters can have long-term detrimental effects on motor development in children.	8/9
Guo et al., 2020 [49]	1998 Yangtze River flood	China	Cohort study	Questionnaire: Denver DST, Gesell DI, WISC-CR, Adaptive Scale of Infant and Children	108.175 mother–child dyad (td = 26.069; preconception exposed = 24.424; prenatal exposed = 28.821; postpartum exp = 29.061)	6.42 ± 1.19 months old	NR	- This study found that prenatal exposure to the Yangtze River flood significantly increased the risk of cognitive impairment in children (*p* < 0.001). - The risk was particularly high when exposure occurred during the first trimester of pregnancy (*p* < 0.001). - The severity and duration of flood exposure were also critical factors, with a greater risk observed for those exposed for longer periods and in more severely affected areas (*p* < 0.001).	Prenatal exposure to natural disasters like floods can lead to long-term cognitive impairments in children.	9/9
Hanc et al., 2022 [50]	Cyclone Aila 2009	India	Cohort study	Questionnaire: Conner’s TRS-R, HRSS	837 mother–child dyad (prenatal exposed *n* = 336; postnatal exposed *n* = 216; td *n* = 285)	8.57 ± 0.72 months old	NR	- The ADHD symptoms were higher in both exposed groups compared to the control group, with more pronounced symptoms in those who were postnatally exposed (*p* < 0.05). - The timing of exposure (prenatal vs. postnatal) and the sex of the children significantly moderated the effect of disaster exposure on ADHD symptoms.	Exposure to severe stressors like natural disasters during prenatal and early postnatal periods can lead to an increased risk of developing ADHD symptoms in children.	9/9
King et al., 2015 [51]	Queensland flood 2011	Australia	Prospective cohort study	Questionnaire: EPDS, STAI, IES-R, PSI, ASQ, ASRS, CBCL	167 mother–child dyads	The children were assessed at different ages: 6 weeks old, 6 months old, 16 months old, 2.5 years old and 4 years old	NR	- Children exposed to the flood during early pregnancy (first trimester) are at greater risk for developmental delays in cognitive functions. - Postnatal follow-ups indicated that prenatal stress negatively affected children’s motor skills.	Children exposed to PNMS tend to lag behind their peers in skills such as language and problem solving. These children may experience delays in both fine and gross motor abilities.	7/9
Kinney et al., 2008 [52]	Hurricanes and tropical storms in Louisiana between 1980 and 1995	USA	Natural experiment studies	DSM-III-R and DSM-IV criteria’s	320.686 mother–child dyad (TD and exposed group numbers are NR)	NR	NR	- This study found that the prevalence of autism disorder (AD) increased in a dose–response manner with the severity of prenatal storm exposure, especially for children exposed during mid to late gestation (*p* < 0.001). - The highest autism prevalence rates were observed in children exposed to severe hurricanes during the middle (5–6 months) or end (9–10 months) of gestation (*p* < 0.001).	Prenatal exposure to severe environmental stressors like hurricanes and tropical storms may increase the risk of autism, particularly when the exposure occurs during sensitive periods of gestation.	Fair
Laplante et al., 2004 [53]	Quebec Ice Storm 1998	Canada	Natural experiment studies	Questionnaire: Bayley Scales of Infant Development MDI, MCDI	58 mother–child dyads	2 years old (SD; NR)	1st trimester *n* = 21; 2nd trimester *n* = 14; 3rd trimester *n* = 23	- Higher levels of prenatal maternal stress (PNMS) due to the ice storm were associated with lower general intellectual and language abilities in toddlers (*p* < 0.01). - The level of PNMS accounted for a significant proportion of variance in the Bayley MDI scores, productive language, and receptive language abilities of toddlers (*p* < 0.05). - The effects were particularly pronounced for those exposed to stress during the first or second trimesters (*p* < 0.01).	Prenatal exposure to high levels of stress can negatively affect a child’s cognitive and language development. The timing of exposure is critical, with early pregnancy exposure leading to more significant impacts.	Fair
Laplante et al., 2007 [42]	Quebec Ice Storm 1998	Canada	Natural experiment studies	Questionnaire: Bayley Scales of Infant Development MDI, IES-R, General health and postnatal depression scales	52 mother–child dyads	25.6 ± 0.9 months old	1st trimester *n* = 19; 2nd trimester *n* = 14; 3rd trimester *n* = 19	- This study found that toddlers exposed to high levels of prenatal maternal stress (PNMS) exhibited less functional (*p* < 0.001) and more stereotypical toy play (*p* < 0.01), with less diversity, compared to toddlers exposed to low levels of PNMS. - The effects were most pronounced for children exposed during the first and second trimesters (*p* < 0.05).	High levels of prenatal maternal stress due to a natural disaster were associated with reduced functional play and increased stereotypical play in toddlers, particularly for those exposed during the early and mid-pregnancy periods.	Fair
Laplante et al., 2008 [54]	Quebec Ice Storm 1998	Canada	Natural experiment studies	Questionnaire: WPPSI-R, PPVT-R	89 mother–child dyads	5.6 ± 0.1 months old	1st trimester *n* = 35; 2nd trimester *n* = 27; 3rd trimester *n* = 27	- This study found that children exposed to high levels of objective prenatal maternal stress (PNMS) due to the ice storm had lower Full-Scale IQs, Verbal IQs, and language abilities compared to those exposed to low or moderate levels of stress (*p* < 0.05).	High levels of objective prenatal maternal stress are associated with lower cognitive and language abilities in children.	Fair
Laplante et al., 2018 [55]	Iowa Flood 2008	USA	Prospective cohort study	Questionnaire: Bayley Scales of Infant and Toddler Development—3rd Edition (Bayley III), MCDI	132 mother–child dyads	30.7 ± 1.0 months old	1st trimester *n* = 38; 2nd trimester *n* = 50; 3rd trimester *n* = 44)	- Higher levels of subjective maternal distress during early pregnancy were associated with lower cognitive functioning in toddlers. - Additionally, there was a significant interaction between subjective distress and the timing of exposure, with the most significant effects observed in those exposed to maternal stress in the first 121 days of pregnancy (*p* < 0.05). - In contrast, boys exposed to higher levels of maternal objective hardship demonstrated greater receptive and productive language abilities compared to girls (*p* = 0.005).	Both subjective distress and objective hardship during prenatal development affect cognitive and language outcomes in children, with specific patterns influenced by the timing of exposure and the child’s sex.	7/9
Lequertier et al., 2019 [33]	Queensland flood 2011	Australia	Natural experiment studies	Questionnaire: BITSEA, IES-R, QFOSS, LES, DASS-21	125 mother–child dyads	16 ± 0.78 months old	1st trimester *n* = 45; 2nd trimester *n* = 45; 3rd trimester *n* = 35	- This study found that greater maternal posttraumatic stress (PTS) symptoms were associated with reduced infant competence (*p* < 0.05). - Boys had significantly more behavioral problems than girls at higher levels of maternal objective hardship and PTS (*p* < 0.05).	Exposure to prenatal maternal stress due to a natural disaster can have domain-specific effects on infant development, particularly in terms of behavioral problems and social-emotional competence.	Fair
Li et al., 2018 [56]	Sand and dust storm in China	China	Quasi experimental study	Questionnaire: PTSD Checklist, PCL-C, DST	1236 children: For the analysis of math test scores; 2693 children: For the analysis of word recognition test scores; 1951 children: For the analysis of counting and speaking ages	NR	NR	- Prenatal exposure to sand and dust storms was associated with negative effects on children’s cognitive function. Every 10 additional days of prenatal exposure led to a 0.20 SD reduction in word test scores, delays in speaking in sentences and counting from one to ten (*p* < 0.01). - The effects were most marked during the sixth and seventh months of gestation (*p* < 0.05).	Prenatal exposure to sand and dust storms has long-term adverse effects on children’s cognitive development, particularly when exposure occurs during critical periods of fetal brain development, such as the sixth and seventh months of gestation.	Fair
McLean et al., 2018 [34]	Queensland flood 2011	Australia	Prospective cohort study	Questionnaire: SPAS, CBCL, C-TRF, IES-R, PDI-Q, PDEQ, DASS, STAI, QFOSS	230 mother–child dyads	48.8 ± 1.3 months old	NR	- Greater objective hardship due to the Queensland floods during pregnancy was significantly associated with greater anxiety symptoms in children at age 4 (*p* < 0.05). - Early exposure to the flood’s during pregnancy was also linked to higher anxiety symptoms (*p* < 0.05). - There were no significant moderating effects of child sex or timing of exposure on these associations (*p* > 0.05).	Prenatal exposure to stress due to natural disasters may have programming effects on childhood anxiety symptoms, emphasizing the importance of understanding both objective and subjective aspects of prenatal maternal stress.	7/9
McLean et al., 2019 [35]	Queensland flood 2011	Australia	Prospective cohort study	Questionnaire: SPAS, CBCL, C-TRF, IES-R, PDI-Q, PDEQ, DASS, STAI, QFOSS	104 mother–child dyads	48.8 ± 1.3 months old	NR	- This study found that the temperamental characteristics of toddlers, such as negative reactivity and shy-inhibited behavior, mediated the relationship between prenatal maternal stress (PNMS) and later childhood anxiety symptoms (*p* < 0.01).	The severity of maternal objective hardship during pregnancy was associated with greater internalizing behaviors in children, mediated by the toddlers’ negative reactivity levels.	7/9
McLean et al., 2021 [36]	Queensland flood 2011	Australia	Prospective cohort study	Questionnaire: SPAS, CBCL, C-TRF, IES-R, PDI-Q, PDEQ, DASS, STAI, QFOSS	4 years old kids *n* = 109; 6 years old kids *n* = 124	6.42 ± 1.19 months old	NR	- This study found no significant associations between disaster-related prenatal maternal stress (PNMS) and childhood anxiety symptoms at 6 years (*p* > 0.05). - Maternal parenting behaviors (overinvolvement and negativity) did not moderate the effects of PNMS on anxiety symptoms (*p* > 0.05). - Poorer maternal concurrent mood at 6 years was associated with greater anxiety symptoms in children, accounting for 26% of the variance (*p* < 0.001).	Maternal mood, rather than prenatal stress exposure or anxiety-maintaining parenting behaviors, is related to anxiety symptoms in school-age children.	7/9
Morales et al., 2023 [37]	Chilian Earthquake 2010	Chile	Quasi experimental study	Questionnaire: CBCL 1.5-5	1549 mother–child dyads (*n* = 933 TD; *n* = 616 treatment group)	18–35 months old (SD; NR)	NR	- Children exposed to the 2010 Chilean earthquake in utero had significantly higher scores on several behavioral and emotional problems, such as emotional reactivity, anxious/depressed, sleep problems, attention problems, and aggression, compared to those not exposed (*p* < 0.05).	Prenatal exposure to natural disasters, such as earthquakes, is associated with increased behavioral and emotional problems in early childhood.	Fair
Moss et al., 2017 [57]	Queensland flood 2011	Australia	Prospective cohort study	Questionnaire: Bayley Scales of Infant and Toddler Development—3rd Edition (Bayley III), QFOSS, IES-R, PDI, PDEQ	145 mother–child dyads	16.48 ± 0.57 months old	1st trimester *n* = 56; 2nd trimester *n* = 53; 3rd trimester *n* = 36	- This study found that flood exposure during pregnancy negatively impacted child cognitive and motor development at 16 months (*p* < 0.05). - Maternal PTSD symptoms and negative cognitive appraisal of the flood were associated with poorer child motor development, with these relationships moderated by the timing of exposure (*p* < 0.05).	Both objective and subjective maternal stress reactions during a disaster can influence child development outcomes, emphasizing the need for comprehensive support and interventions targeting maternal mental health during pregnancy in the context of natural disasters.	7/9
Moss et al., 2018 [58]	Queensland flood 2011	Australia	Prospective cohort study	Questionnaire: Bayley Scales of Infant and Toddler Development—3rd Edition (Bayley III), QFOSS, IES-R, PDI, PDEQ	150 mother–child dyads	30.26 ± 0.72 months old	1st trimester *n* = 63; 2nd trimester *n* = 55; 3rd trimester *n* = 32	- Severe maternal PTSD symptoms predicted poorer child fine motor development, and maternal peritraumatic distress predicted better development (*p* < 0.05). - The impact of flood-related variables on child development was found to vary depending on the timing of exposure during gestation, with later exposure predicting greater improvements in gross motor development between 16 and 30 months of age (*p* < 0.05).	Different types of maternal stress reactions to a natural disaster during pregnancy can affect child development outcomes, particularly in cognitive and motor domains, and that these effects can vary over time depending on the type and timing of exposure.	7/9
Nomura et al., 2019 [38]	Hurricane Sandy 2012	USA	Quasi experimental study	Questionnaire: EPDS, IBQ-R, Traumatic Exposure Instrument	310 mother–child dyads	6 months old (SD; NR)	1st trimester *n* = 75; 2nd trimester *n* = 20; 3rd trimester *n* = 15; before disaster birth *n* = 200)	- This study found that prenatal maternal depression was associated with lower emotion regulation and greater distress in infants (*p* < 0.05). - These adverse effects were amplified by in utero exposure to Superstorm Sandy, particularly in terms of increased activity, distress, sadness, and reduced cuddliness (*p* < 0.05).	The combination of prenatal maternal depression and exposure to a natural disaster can have a magnified impact on infant temperament, including greater activity, distress, and sadness.	Good
Nomura et al., 2023 [39]	Hurricane Sandy 2012	USA	Prospective cohort study	Questionnaire: PAPA, Structured diagnostic interviews	163 mother–child dyads (*n* = 66 in utero exposed; *n* = 97 nonexposed)	3.19 years old (SD:NR)	1st trimester *n* = 13; 2nd trimester *n* = 19; 3rd trimester *n* = 34	- In utero exposure to Superstorm Sandy was significantly associated with an increased risk of various psychiatric disorders in preschool children, including anxiety disorders, depressive disorders, and attention-deficit/disruptive behavioral disorders (*p* < 0.05). - The risks varied by sex, with males more likely to develop attention-deficit/disruptive behavioral disorders and females more prone to anxiety and depressive disorders (*p* < 0.05).	Prenatal exposure to a major weather-related disaster can have long-term mental health consequences for children, with distinct patterns of psychopathological outcomes based on the child’s sex.	7/9
Simcock et al., 2016 [59]	Queensland flood 2011	Australia	Prospective cohort study	Questionnaire: QFOSS, IES-R, PDI, PDEQ	2 months age *n* = 106; 6 months age *n* = 115; 16 months age *n* = 130	2 months old = 1.99 ± 0.42 months old; 6 months old = 6.25 ± 0.33 months old; 16 months old = 16.11 ± 0.77 months old	1st trimester *n* = 58; 2nd trimester *n* = 48; 3rd trimester *n* = 19	- At 2 months, higher levels of PNMS were positively related to motor development, but at 6 and 16 months, a negative association was observed, especially when flood exposure occurred later in pregnancy or when mothers had negative cognitive appraisals of the event (*p* < 0.05).	The timing of prenatal exposure to stress and the mother’s perception of the event significantly impacts the infant’s motor development at different stages of early infancy.	7/9
Simcock et al., 2018 [60]	Queensland flood 2011	Australia	Prospective cohort study	Questionnaire: ASQ-3	2 months age *n* = 106; 6 months age *n* = 115; 16 months age *n* = 129; 2.5 years old *n* = 124; 4 years old *n* = 113	2 months old = 1.99 ± 0.42 months; 6 months old = 6.25 ± 0.33 months; 16 months old = 16.11 ± 0.77 months; 2.5 years old = 29.98 ± 0.71; 4 years old = 48.65 ± 0.91	NR	- High objective flood exposure or a negative appraisal, especially in later pregnancy, predicted poorer gross motor skills which rapidly improved across early childhood (*p* < 0.05). - Fine motor skill development was influenced by the child’s sex, with improvements observed in girls over time but not boys (*p* < 0.05).	Stress in pregnancy has a long-term impact on children’s motor development, particularly gross motor skills.	7/9
Simcock et al., 2019 [40]	Queensland flood 2011	Australia	Prospective cohort study	Questionnaire: CBCL, QFOSS, IES-R, PDI, PDEQ	2.5 years old *n* = 134; 4 years old *n* = 118	2.5 years old = 30.25 ± 1.44 months old; 4 years old = 48.80 ± 1.29 months old	NR	- Severe objective flood-related hardship in pregnancy predicted higher sleep problem scores at 2.5 years, and a negative maternal cognitive appraisal of the flood predicted lower attention problem scores at 2.5 years (*p* < 0.05). - The cross-lagged panel analysis showed that anxious/depressed symptoms at age 2.5 predicted sleep problem scores at age 4 (*p* = 0.001).	Prenatal maternal stress due to a natural disaster can have long-term effects on child development, specifically in sleep, attention, and mood problems.	7/9
Walder et al., 2014 [61]	Quebec Ice Storm 1998	Canada	Prospective cohort study	Questionnaire: ASSQ, IES-R, Storm 32 Scale	89 mother–child dyads (*n* = 46 girls; *n* = 43 boys)	6.5 years old (SD:NR)	1st trimester *n* = 28; 2nd trimester *n* = 33; 3rd trimester *n* = 28	- Greater objective and subjective prenatal maternal stress (PNMS) predicted higher autism spectrum screening questionnaire (ASSQ) scores among children at age 6.5 (*p* < 0.05). - Objective stress had a more significant impact on children exposed during the first trimester (*p* < 0.05).	Prenatal exposure to severe maternal stress from a natural disaster, especially during early pregnancy, can significantly influence the development of autism-like traits in children.	7/9
Zhang et al., 2018 [41]	Hurricane Sandy 2012	USA	Prospective cohort study	Questionnaire: IBQ-R, ECBQ, PDS, EPDS	318 mother–child dyads	6–24 months old children (SD: NR)	NR	- Both objective exposure to Superstorm Sandy and subjective stress reactions in utero predicted developmental trajectories of temperament in early childhood (*p* < 0.05). - Children exposed to Sandy in utero showed significant changes in temperament, such as increased activity level but decreased high-intensity pleasure, approach, and fearfulness over time (*p* < 0.05).	Both objective and subjective prenatal maternal stress due to natural disasters can have long-term impacts on the developmental trajectories of child temperament.	7/9

ASQ: Ages and Stages Questionnaire, ASRS: Adult ADHD Self-Report Scale, Bayley Scales of Infant Development, BDI: Beck Depression Inventory, BITSEA: Brief Infant–Toddler Social and Emotional Assessment, BOT-2: Bruininks–Oseretsky Test of Motor Proficiency, Second Edition, BOTMP: Bruininks–Oseretsky Test of Motor Proficiency, CBCL: Child Behavior Checklist, Conner’s TRS-R: Conners’ Teacher Rating Scale—Revised, DASS-21: Depression Anxiety Stress Scales-21, DST: Developmental Screening Test, ECBQ: Early Childhood Behavior Questionnaire, EPDS: Edinburgh Postnatal Depression Scale, Gesell DI: Gesell Developmental Inventory, HRSS: Holmes–Rahe Stress Scale, IBQ-R: Infant Behavior Questionnaire—Revised, IES-R: Impact of Event Scale—Revised, LES: Life Events Scale, QFOSS: Queensland Flood Objective Stress Scale, MCDI: MacArthur–Bates Communicative Development Inventories, MCDI-III: MacArthur–Bates Communicative Development Inventories-III, MDI: Mental Development Index, PAPA: Preschool Age Psychiatric Assessment, PDEQ: Peritraumatic Dissociative Experiences Questionnaire, PDI: Peritraumatic Distress Inventory, PPVT-R: Peabody Picture Vocabulary Test—Revised, PSI: Parenting Stress Index, PTSD Checklist PCL-C: Post Traumatic Stress Disorder Checklist—Civilian Version, SPAS: Spence Preschool Anxiety Scale, STAI: State-Trait Anxiety Inventory, TADI: Teacher’s ADHD Rating Scale, VMI: Visual–Motor Integration, WISC-CR: Wechsler Intelligence Scale for Children—Chinese Revised, WPPSI-R: Wechsler Preschool and Primary Scale of Intelligence—Revised, SD: Standard Deviation, NR: Not Reported. * Quality of each cohort and cross-sectional study was assess using the Newcastle–Ottawa Quality Assessment Scale (NOQAS). Quality of each natural experiment and quasi-experimental study was assess using TREND Statement Checklist [31].

## 4. Discussion

The aim of this study was to investigate the neurodevelopmental effects in children exposed to natural disasters during the intrauterine period. In addition, the effects of gestational age and sex of the child on these neurodevelopmental changes were assessed.

As a result, the present systematic review of the literature reveals that children of pregnant women exposed to natural disasters are affected in a multitude of ways. Based on the findings of the studies, it can be concluded that natural disasters have a negative impact on cognitive development, language development, autism/autism-like characteristics, motor skills, performance in mathematics, mental development, sleep problems, attention problems, behavioral and emotional problems, and various psychiatric comorbidities in children. Although the number of studies on the effect of time of exposure is limited and the results are inconsistent, these inconsistencies can be attributed to methodological factors such as the lack of studies on PNMS related to natural disasters and the type of stress exposure. However, it can be concluded that exposure to natural disasters in the first trimester brings a higher risk according to the studies examined. Despite mounting evidence linking PNMS with neurodevelopmental disorders (including subclinical presentations), several limitations remain. These include the retrospective design of many studies and the fact that the clinical evaluation of children was performed many years after the natural disaster. This has negative implications for early intervention.

Among the reviewed studies, the impact of floods has been analyzed the most. In 2023, floods were the most frequent natural disaster in the world, followed by storms [62]. Floods are particularly common in Asia, Africa, and some European countries, affecting millions of people annually. The frequency and severity of flood events are increasing due to climate change [63].

In respect to the instruments used to assess children’s neurodevelopment, different scales were used in the reviewed studies. Researchers often used the CBCL 1.5-5 version to assess the social and emotional effects of natural disasters on children, and seven studies used this scale [32,34,35,36,37,40,51]. The CBCL 1.5-5 is a screening tool used for the assessment of multiple behavioral and emotional difficulties in children aged 1.5 to 5 years. This version includes seven subscales: emotionally reactive (9 items), anxious/depressed (8 items), somatic complaints (11 items), withdrawn (8 items), sleep problems (7 items), attention problems (5 items), and aggression (19 items). Items are scored on a three-point Likert scale ranging from 0 (not true) to 2 (very or often true), with higher scores indicating more problematic behaviors. The Bayley Scales of Infant and Toddler Development—third edition (BSID-III), another scale that assesses cognitive, social, and emotional abilities and is often used in studies, was used in six studies [42,43,53,55,57,58]. The Bayley is a standardized assessment of developmental functioning and is widely used in studies of child development. The BSID-III has high internal consistency and has been extensively validated with mean reliability coefficients of 0.91 for the Cognitive Scale, 0.86 for the Fine Motor Scale, and 0.91 for the Gross Motor Scale [64]. The Bayley Scales are useful scales that have been used for several decades to detect early developmental delays in clinical practice and research [65]. Moss et al. [57] assessed the cognitive and motor outcomes of flood exposure during pregnancy using the BSID-III scale, while Morales et al. [37] assessed the outcomes of emotional reactivity, anxious/depressed, sleep problems, attention problems, and aggression in children of mothers exposed to earthquakes during pregnancy using the CBCL 1.5-5 scale. The BSID-III and CBCL 1.5-5 are complementary tools used for different purposes. While the Bayley-III assesses children’s general development, the CBCL is more of a tool for understanding children’s emotional and behavioral functioning. The two scales can be used together, particularly in cases where developmental delays and mental health problems need to be addressed together.

Pregnancy is an important period when environmental exposures, such as maternal psychological stress, have lifelong consequences for the developing infant, a concept termed ‘fetal programming’ [66]. The effects of maternal psychosocial stress on fetal development and later life health and disease are not the result of a single pathway but are mediated by multiple stress transmission mechanisms (maternal cortisol, catecholamines, cytokines, reactive oxygen radicals, serotonin/tryptophan, microbiota) acting together in a synergistic manner [66]. Cortisol is a glucocorticoid hormone released by stimulation of the hypothalamic–pituitary–adrenal (HPA) axis in response to stressors [67]. The role of cortisol as a mediator of prenatal stress has been extensively investigated in recent years and has previously been reviewed in detail [68,69,70,71]. In response to acute or chronic psychological stress, the maternal HPAA is activated by higher brain structures and, as a result, the adrenal cortex synthesizes and releases cortisol into the maternal circulation [72]. After release into the maternal circulation, the highly lipophilic cortisol crosses the placental barrier and reaches the fetus. However, the placental enzyme 11beta-hydroxysteroid dehydrogenase type 2 (11β-HSD-2) inactivates approximately 80–90% of maternal cortisol. In addition, fetal HPAA does not produce cortisol until late in pregnancy, suggesting that the fetus is completely dependent on maternal cortisol for most of pregnancy. The intrauterine determination of fetal HPAA activity later in life is thought to be one of the key mechanisms of ‘fetal programming’ [66]. This mechanism may explain the neurodevelopmental effects of prenatal exposure to natural disasters through acute maternal stress.

The placental enzyme 11beta-hydroxysteroid dehydrogenase type 2 (11β-HSD-2) inactivates about 80–90% of maternal cortisol. As a result, fetal plasma concentrations are 5–10 times lower than maternal levels under physiological conditions [73]. This mechanism is thought to protect the fetus from excessive maternal cortisol concentrations under physiological conditions. Prenatal stress itself, or maternal anxiety as a marker of prenatal stress, has been shown to reduce 11β-HSD-2 expression and activity in humans and animal models [73,74,75,76]. This may explain why exposure to natural disasters in the first trimester leads to worse neurodevelopmental outcomes in children in most of the studies analyzed. On the other hand, human placental 11β-HSD-2 activity decreases near delivery in uncomplicated pregnancies [77], resulting in greater maternal cortisol transmission to the fetus in late pregnancy. This may be the mechanism underlying the greater neurodevelopmental effects of late-gestational exposure to natural disasters in children in some of the studies reviewed. Information on the relationship between exposure to natural disasters and gestational age is still limited, and more studies are needed.

A review published in 2022 [67] examined the relationship between stress in pregnant women and fetal sex and found that female fetuses had a less accelerated fetal heart rate [78] and lower placental 11b-hydroxysteroid dehydrogenase type 2 function [79] compared to male fetuses. Less 11b-hydroxysteroid dehydrogenase type 2 means that the fetus is exposed to more maternal cortisol. Higher levels of cortisol in the female fetus may be a factor associated with more reading and language problems in girls in the studies we reviewed. Most of the studies in this review did not include a gender perspective on the effects of prenatal stress. There is evidence that maternal stress affects male and female newborns differently [80]. Future studies addressing natural disasters as a factor in PNMS should consider fetal sex and analyze how prenatal stress affects infant outcomes from a gender perspective.

### 4.1. Limitations

The small number of databases consulted and the lack of a meta-analytic perspective are the main limitations of this study. Databases such as PsycINFO, LILACS, Latindex, and DergiPark could have been included for further review, allowing for a more comprehensive understanding of studies from diverse regions and enhancing the likelihood of identifying relevant research across various cultural and regional contexts. Secondly, the generalizability of the results is difficult due to the heterogeneity in the methodology (differences in sample size, scales used, statistical methods used, etc.) of the studies reviewed.

### 4.2. Strengths

The main strength is that we carried out a scoping review in accordance with the PRISMA extension for scoping reviews guidelines. Our research was a comprehensive study covering all types of natural disasters and analyzing all neurodevelopmental effects in children. In addition, the quality of the analyzed studies was mostly high, which is an important indicator of the reliability of the results of these studies.

### 4.3. Clinical and Research Implications for Future Research

Is exposure to a natural disaster alone a predictor of adverse mental health? Furthermore, to what extent do the death of relatives and various unfavorable conditions experienced after the disaster increase this risk? With regard to the earthquake natural disaster, it is unclear whether the duration of stay in the region after the disaster poses an additional teratogenic risk in terms of exposure to chemicals. In particular, the relationship between the mental health outcomes of early children exposed to these events in the intrauterine period remains unclear. Follow-up studies established for early childhood allow for the effects of objective and subjective components of maternal exposure to be assessed and separated.

A new study has been initiated to address these questions and minimize the limitations of previous research. This study is currently in the design phase and is planned to be conducted in the future. The aim is to increase knowledge on this subject and contribute to the literature by using a prospective research design to determine the extent to which postnatal maternal stress caused by a major natural disaster predicts autism spectrum disorder (ASD) symptoms in early childhood. In addition, the effects of child gender and timing of exposure, as well as the effects of other maternal and child factors on PNMS and autism-like features, including social, emotional, and cognitive features, will be evaluated.

## Figures and Tables

**Figure 1 behavsci-14-01054-f001:**
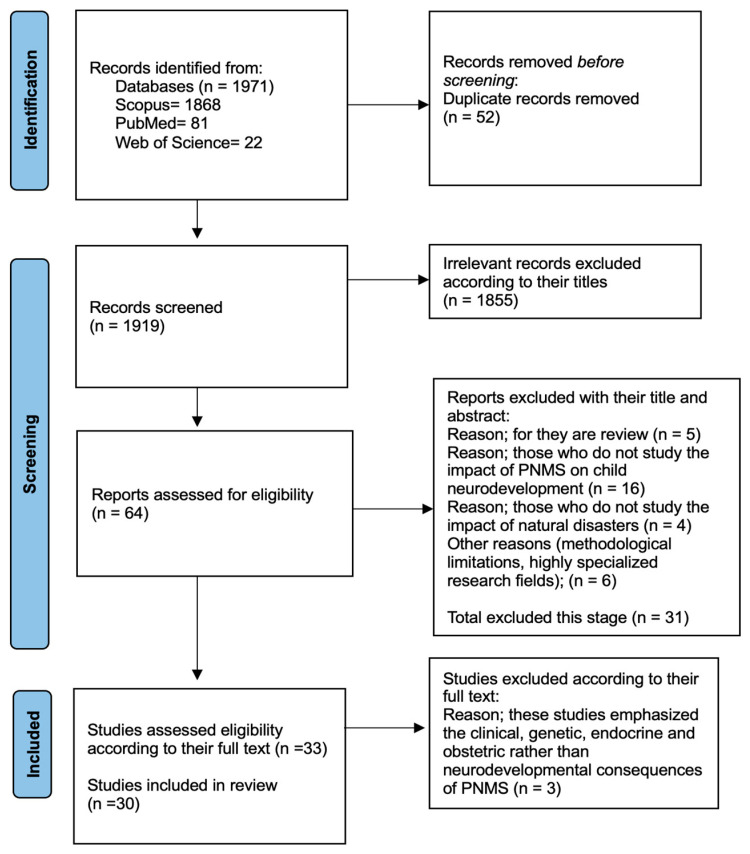
PRISMA 2020 [29] flow diagram.

**Table 1 behavsci-14-01054-t001:** Inclusion and exclusion criteria.

**Inclusion Criteria**
**Population:** Neonates and infants exposed to natural disasters before birth
**Language:** English, Spanish, Turkish
**Year of publication:** No time restriction
**Design:** Observational, experimental, cohort, longitudinal, quasi-experimental
**Exclusion Criteria**
**Design:** Literature reviews, meta-analyses, editorials
**Population:** Pregnant women exposed to maternal stress (different from a natural disaster)

## Data Availability

The original contributions presented in this study are included in the article. Further inquiries can be directed to the corresponding author.

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
