# Peer review of "Natural Disasters as a Maternal Prenatal Stressor and Children’s Neurodevelopment: A Systematic Review"

_behavsci, 2024, doi:10.3390/bs14111054_

Round 1
Reviewer 1 Report
Comments and Suggestions for Authors
Great review! Minor comments:
-Line 36, says natural disasters affect everyone equally. May try to rephrase to highlight that people may not actually be affected equally when exposed to natural disaster, especially considering severity and other socioeconomic risk factors (ie poverty, housing security)
-line 141 missing verb
Reviewer 2 Report
Comments and Suggestions for Authors
This article presents an interesting review about the influence of prenatal maternal stress and the post-partum neurodevelopmental status of children. Overall, the article is very clear and well written. I have only a few minor comments that I believe should be addressed before the article can be deemed acceptable for publication.
Major Issues
-
In the abstract it is reported that “The intrauterine period is a time of high sensitivity in the development of the embryo and 14 the fetus. Therefore, high levels of maternal stress are closely associated with healthy brain development in the neonatal and early childhood periods” Shouldn’t this be low levels of maternal stress? I would suggest the authors to check the accuracy of the abstract.
-
Line 115/Figure 1. There is a lack of clarity on why some studies have been excluded. In particular, what are “other reasons” for excluding articles in the “Reports assessed for eligibility” section? I would suggest the authors to add more details about the procedure in the text or in the figure.
-
Figure 1. It is also not clear how the authors passed from 33 studies assessed for eligibility to 30 included in review. What are the reasons for which the three articles have been discarded? Authors should expand on this.
-
Line 120. It is not clear whether both the authors reviewed all the studies or if they coded a subset of the articles each. If the latter, a measure of interindividual reliability should be provided (e.g. on common coding of a small subset of articles).
-
Table 2. It is not clear what the arrow up symbol is used for in the Table. A description of its meaning should be provided. I would suggest the authors to add it in the table’s caption.
-
As a limitation, the authors indicated that “the small number of databases consulted [...] are the main limitations of this study”. What other databases could the authors have used? I would suggest to add a note of a possible larger database between parentheses.
Minor Issues
-
Line 141. “Eleven of reviewed studies [31-41] that maternal” is there a missing word after the references? The sentence seems to be incomplete.
-
On page 6 it is reported that “In their studies, researchers have not only examined the neurodevelopmental effects of exposure to natural disasters in children, but also whether the period of pregnancy during which exposure to natural disasters occurs is effective or not” Does this refer to all the included studies? Or only to a subset of them. I would suggest clarifying which studies are the authors referring to.
-
Line 394. “A study”. Is this referring to the current work or to a future already initiated work? This is not very clear from the sentence.
